# Tissue Hypoxia and Associated Innate Immune Factors in Experimental Autoimmune Optic Neuritis

**DOI:** 10.3390/ijms25053077

**Published:** 2024-03-06

**Authors:** Zhiyuan Yang, Cristina Marcoci, Hatice Kübra Öztürk, Eleni Giama, Ayse Gertrude Yenicelik, Ondřej Slanař, Christopher Linington, Roshni Desai, Kenneth J. Smith

**Affiliations:** 1Department of Neuroinflammation, UCL Queen Square Institute of Neurology, University College London, London WC1N 1PJ, UKk.ozturk@ucl.ac.uk (H.K.Ö.);; 2Institute of Pharmacology, First Faculty of Medicine, Charles University and General University Hospital in Prague, 12800 Prague, Czech Republic; 3School of Infection and Immunity, The Sir Graeme Davies Building, Glasgow G12 8TA, UK

**Keywords:** oxidative stress, superoxide, nitric oxide, peroxynitrite, hypoxia inducible factor-1α, multiple sclerosis

## Abstract

Visual loss in acute optic neuritis is typically attributed to axonal conduction block due to inflammatory demyelination, but the mechanisms remain unclear. Recent research has highlighted tissue hypoxia as an important cause of neurological deficits and tissue damage in both multiple sclerosis (MS) and experimental autoimmune encephalomyelitis (EAE) and, here, we examine whether the optic nerves are hypoxic in experimental optic neuritis induced in Dark Agouti rats. At both the first and second peaks of disease expression, inflamed optic nerves labelled significantly for tissue hypoxia (namely, positive for hypoxia inducible factor-1α (HIF1α) and intravenously administered pimonidazole). Acutely inflamed nerves were also labelled significantly for innate markers of oxidative and nitrative stress and damage, including superoxide, nitric oxide and 3-nitrotyrosine. The density and diameter of capillaries were also increased. We conclude that in acute optic neuritis, the optic nerves are hypoxic and come under oxidative and nitrative stress and damage. Tissue hypoxia can cause mitochondrial failure and thus explains visual loss due to axonal conduction block. Tissue hypoxia can also induce a damaging oxidative and nitrative environment. The findings indicate that treatment to prevent tissue hypoxia in acute optic neuritis may help to restore vision and protect from damaging reactive oxygen and nitrogen species.

## 1. Introduction

Optic neuritis is an inflammatory demyelinating disease of the optic nerve that can cause severe visual deficits, including blindness [1,2]. The deficits often appear to resolve over time, but more permanent residual deficits can frequently be revealed upon careful examination [2,3,4,5]. The reversible and permanent visual deficits are typically attributed respectively to the inflammatory demyelination and neuroaxonal degeneration that accompany them [6,7,8,9]. This damage appears to involve a range of innate immune mechanisms, including the actions of reactive oxygen species (ROS) [10,11,12,13,14,15,16,17], nitric oxide [17,18,19,20,21,22,23,24,25,26,27,28], peroxynitrite [19,21,23,29,30,31,32] and mitochondrial failure [17,33,34,35,36,37,38,39], reviewed in [14,40,41,42,43,44,45], all of which are increased in optic neuritis [11,12,46,47,48]. Accordingly, treatments to control ROS have been found to provide protection in experimental autoimmune optic neuritis [46,49,50,51], which is a common feature of the animal model of multiple sclerosis, experimental autoimmune encephalomyelitis (EAE) [52,53].

There is no satisfactory treatment for optic neuritis. Steroids can shorten the duration of the attack, but they barely change the long-term outcome [4,9,54,55,56,57]. A treatment avenue would arise if it was possible to control the innate immune factors involved in the damage, but their control is complicated by the fact that the cause of their initiation is unclear. The innate immune factors are common features of inflammatory lesions, but, for example, ROS and mitochondrial dysfunction can both precede the histological appearance of inflammation by a week or more [10,11,17,52], suggesting that they are not caused by inflammation, at least if inflammation is defined by the arrival and presence of inflammatory cells. The innate mechanisms are, however, all promoted in tissue if the tissue is hypoxic, i.e., if the tissue has a pathologically low concentration of oxygen. Thus, hypoxia is a potent promotor of mitochondrially produced superoxide and the consequent cascade of reactive oxygen species [58,59,60,61]. Hypoxia also induces the production of nitric oxide via the formation of the inducible form of nitric oxide synthase (iNOS) [62,63,64], namely, the form of the enzyme that produces large amounts of nitric oxide continuously. The free radicals superoxide and nitric oxide combine avidly to form the highly toxic oxidising agent peroxynitrite [65,66], which damages proteins by nitrating tyrosine to nitrotyrosine [65,67]. Hypoxia also, of course, impairs mitochondrial function, and such damage is prominent in EAE [16,37]. Hypoxia therefore emerges as a potential cause of the innate immune factors believed to be responsible for demyelination and axonal degeneration in optic neuritis, but it is not yet known whether the inflamed optic nerve is hypoxic.

There is evidence that the vasculature of the inflamed optic nerve may be hypoperfused as several studies have found deficits in blood flow in the retina and near the optic nerve head in acute optic neuritis in humans [68,69,70,71] and animals [72], and this blood supply is derived from the optic nerve. The hypoperfusion appears to be caused by raised concentrations of the potent vasoconstricting agent, endothelin-1, in the blood and cerebrospinal fluid in both optic neuritis and MS [73,74,75,76,77]. However, it is not inevitable that a reduction in vascular perfusion will result in tissue hypoxia because the reduction may be a physiological response to a reduction in metabolic demand. Such a reduction can occur if, for example, the demand for oxygen is reduced by the inhibition of mitochondrial complex IV by nitric oxide [17,78,79] (i.e., the existence of a condition of virtual hypoxia, also known as histotoxic hypoxia, in which the oxygen concentration can actually rise because of reduced oxygen utilisation). If the optic nerve is *actually* hypoxic (also known as hypoxic hypoxia, where the oxygen concentration is low, e.g., due to reduced blood flow and thus reduced oxygen delivery), the hypoxia may indicate an avenue to treatment because we have found, in neuroinflammatory lesions, that breathing raised oxygen or treatment with vasodilating nimodipine can quite promptly reverse hypoxia. Furthermore, such protection from hypoxia also provides protection from the associated disability, demyelination and degeneration that occur in EAE [80,81,82] and from the pattern III demyelination in experimental white matter inflammation [83]. Notably, if the optic nerve in optic neuritis is actually hypoxic, corrective treatment could simultaneously reduce all the mentioned mediators of innate damage, namely, ROS, nitric oxide, peroxynitrite and mitochondrial failure. Here, we explore whether the optic nerve is actually hypoxic in experimental autoimmune optic neuritis and the relationship of the hypoxia with the presence of the innate immune factors.

## 2. Results

Animals were selected for a histological study of their optic nerves based on the expression of a hindlimb neurological deficit (weakness) because it is not easy to screen animals for their expression of optic neuritis while they are awake. Our previous research has found that in a DA rMOG model of EAE, as examined here, almost 100% of animals will express optic neuritis eventually, but the expression is not necessarily coincident with the expression of a hindlimb deficit. Most commonly, optic neuritis occurs first, but, in some animals, it occurs second. Thus, in this study, optic nerves were studied that expressed optic neuritis at different stages of lesion development.

Twenty-six of the thirty animals immunised with rMOG expressed the hind limb and tail weakness/paralysis expected with animals exhibiting a neurological deficit due to EAE (Table 1). Of the twenty-six animals with hindlimb and tail deficits, twenty-three exhibited bilateral optic neuritis, one had only unilateral disease, and two had no inflammation in their optic nerves. The two animals with no optic neuritis had been perfused at the first peak of hindlimb and tail disease (11 and 14 days post-immunisation, respectively). Thus, in these two animals, the pathology in the spinal cord preceded that in the optic nerves. Of the four animals without obvious hindlimb or tail deficits, one animal (perfused at 29 days post-immunisation) had bilateral and one (perfused at 45 d post-immunisation.) had unilateral optic neuritis upon histology examination: the remaining nerves were not inflamed (perfused at 18 and 44 days post-immunisation., respectively). In total, forty-seven out of fifty-six nerves collected for histological examination were inflamed and designated EAE-ON, and the remaining nine nerves were not inflamed and designated EAE-NON.

Inflammation typically affected the anterior (orbital) portion of the nerve first (Figure 1e), often spreading caudally towards the chiasm, and sometimes involving the chiasm (Figure 1f). Immunofluorescent examination of labelling for IBA1 and ED1 revealed that optic nerves from animals immunised for EAE expressed significantly higher microglial/macrophage density (IBA1, *p* < 0.001; Figure 1b) and activation (ED1, *p* < 0.001; Figure 1d) than optic nerves from control animals ‘immunised’ with IFA alone. Labelling for IBA1 and ED1 was also significantly higher in EAE-ON than in EAE-NON (IBA1: *p* = 0.0011, ED1: *p* = 0.014).

Immunohistochemical labelling for the marker of hypoxia, HIF1α, was significantly higher in EAE-ON nerves than in nerves from IFA animals, regardless of whether the nerves were obtained during the first or second peak of the disease (*p* = 0.0029; Figure 2c). The labelling for HIF1α was more intense in the most inflamed nerves (*p* = 0.033; Figure 2d). The complementary label for hypoxia, pimonidazole, confirmed that the labelling for HIF1α correlated with evidence of low oxygen concentration, and, again, the labelling was significantly more intense in the EAE-ON group compared with the IFA group (*p* = 0.0017; Figure 2e).

The number and diameter of blood vessels were measured in nerves immunolabelled for the endothelial expression of GLUT1. Sections from EAE-ON nerves showed an apparently significant increase in the number of vessels compared with nerves from IFA animals (*p* = 0.0090). The trend towards significance compared with EAE-NON nerves may have been limited by the small sample size of nerves in the EAE-NON group (Figure 3b). The apparent increase in vessel number was accompanied by a significant increase in vessel diameter, compared with either EAE-NON nerves (*p* = 0.0038) or nerves from IFA animals (*p* < 0.001; Figure 3c).

Labelling for iNOS was significantly more intense in the EAE-ON group at the peak of the disease (i.e., 2 days after onset), especially in the more profoundly inflamed lesions (Figure 4c-iii; see also yellow line in Figure 4g) than in the IFA or EAE-NON nerves (EAE-NON: *p* = 0.024, IFA: *p* < 0.001; Figure 4d). However, by day 4 after the onset of disease expression, the labelling for iNOS was reduced in the EAE-ON nerves to resemble that in the non-inflamed nerves, and labelling was significantly lower than that in EAE-ON at the peak of the disease (*p* = 0.0015; Figure 4d, see also Figure 4g). Labelling for 3NT was also significantly more intense in EAE-ON, especially at the peak of the disease, compared with the IFA or EAE-NON nerves (EAE-NON: *p* = 0.040, IFA: *p* = 0.0011; Figure 4f). Labelling for 3NT was absent from the most inflamed nerves (see yellow line in Figure 4h). DHE was only injected into animals perfused on day 4 after the onset of disease expression, i.e., at a time after the labelling for iNOS had reduced, but the findings nonetheless showed significantly more cells labelled after DHE in the EAE-ON group compared with the EAE-NON and IFA nerves (EAE-NON: *p* = 0.14, IFA: *p* = 0.055, EAE-NON and IFA: *p* = 0.031; Figure 4b). Labelling for iNOS and 3NT was most common in the cytoplasm of cells in inflamed optic nerves, but the autofluorescence from DHE was mainly seen in cell nuclei (Figure 4a-iii).

## 3. Discussion

The main finding of this study is that in optic neuritis due to EAE, acutely inflamed optic nerves are significantly hypoxic. The hypoxia is accompanied by evidence of significant increases in the expression of a range of deleterious innate immune factors, including superoxide, nitric oxide, and the potent oxidising agent, peroxynitrite, formed by the combination of the free radicals superoxide and nitric oxide. The increased production of superoxide (evidenced by DHE-induced fluorescence) and nitric oxide (evidenced by the appearance of the iNOS enzyme) were an expected consequence of tissue hypoxia [58,59,60,61,62,63,64], and these free radicals combined avidly to form peroxynitrite [65,66] (evidenced by the appearance of protein damage by labelling for 3NT). The hypoxia and simultaneously raised expression of deleterious innate immune factors raises the possibility that hypoxia may contribute to, or perhaps even be responsible for, the appearance of innate immune factors.

The optic nerves were almost certainly hypoxic because of inadequate vascular perfusion, and this interpretation is supported by observations of reduced vascular perfusion of the retina and optic nerve head in both human [68,69,70,71] and experimental [72] optic neuritis. The cause of hypoperfusion is probably a combination of vasoconstriction, raised tissue pressure (oedema) and the physical compression of the vessels, with the balance of these phenomena varying between individuals and at different stages of disease progression. The vasoconstriction can be attributed, at least in part, to increased plasma and cerebrospinal fluid concentrations of the potent vasoconstricting agent endothelin-1, which is well documented in both optic neuritis and MS [73,74,75,76,77]. A rise in tissue pressure due to oedema resists vascular perfusion because perfusion is driven by a difference in fluid pressure at either end of the vessels, and this pressure gradient into the oedematous tissue is diminished if the tissue pressure is raised. The physical compression of vessels occurs with a rise in tissue pressure, especially if the oedematous tissue swells to become constrained by the inelastic dura or by the bone of the optic foramen. The importance of such swelling is immediately apparent upon direct observation of the inflamed lumbar spinal cord in EAE because the cushion of surrounding cerebrospinal fluid is displaced and the cord immediately abuts the dura, which compresses, indeed flattens, the surface veins [81]. A detailed MRI examination of optic neuritis [8] emphasised the role of inflammation and oedematous ‘pressure block’ as a mechanism underlying visual loss. Localised pressure can indeed block axonal conduction by deforming and depolarising axons [84], but, here, we advance this interpretation by considering that it is vascular insufficiency and resulting tissue hypoxia that is the more likely explanation. Hypoxia readily causes reversible and irreversible conduction failure, depending on the severity and longevity of the hypoxia. Mild and/or brief hypoxia causes mitochondrial failure and a lack of sufficient ATP to maintain the resting potential. In axons, the resulting depolarisation causes conduction failure (and visual loss) by preventing impulse propagation or reducing the ability of axons to conduct impulses at a higher frequency, depending on the severity of the energy crisis.

The results show an apparent increase in vessel number in acute optic neuritis, combined with an increase in vessel diameter (Figure 3). The increase in vessel number is observed quite early in the disease course, and perhaps before the opportunity for significant vessel proliferation. It is therefore possible, if not likely, that the increase in vessel number is apparent rather than real, and due to the increased frequency that dilated vessels will appear in a particular tissue section. Vessel dilation appears to contradict the interpretation of vasoconstriction as a cause of hypoxia, but the dilated vessels are largely the numerous capillaries and the constricted vessels are the few arteries and arterioles.

The finding that the inflamed optic nerve is hypoxic is potentially clinically important because it indicates potential routes to neuroprotective treatment. Routes that overcome hypoxia may not only restore mitochondrial function but also simultaneously reduce all the damaging innate immune markers examined in this study because all are promoted by tissue hypoxia, as noted above. Our previous research has revealed that the prevention of hypoxia in inflamed lesions with either raised inspiratory oxygen or with vasodilating nimodipine is very effective in promptly raising the oxygenation of the lesions, even to normal levels [81]. These treatments are also very effective in restoring neurological function [80,82] and protecting tissue structure (e.g., protection from demyelination) [80,83], but the two treatments are not equivalent in their action. Treatment with oxygen has the advantage of increasing the oxygenation of tissue even if the vessels are too narrowed to allow for the passage of red cells, as long as the vessels have any flow of plasma in which the oxygen is dissolved. However, a potential disadvantage of oxygen treatment is the risk, if vascular perfusion is good, of increasing the oxygenation of the tissue above normal levels and so promoting the production of reactive oxygen species [85]. Treatment with nimodipine has the advantage that it avoids the risk of over-oxygenation because treatment results in the increased perfusion of normally oxygenated blood, rather than hyper-oxygenated blood. However, a disadvantage of nimodipine is that although the drug dilates constricted vessels, it understandably cannot dilate vessels that are narrowed due to physical compression by, for example, oedema. Thus, the choice of treatment is influenced by the mechanisms responsible for causing the tissue hypoxia, such as whether hypoperfusion is due to the constriction or compression of vessels.

Hypoxia of the optic nerve will promote a tissue energy crisis by impairing mitochondrial function. It is therefore relevant that a new approach to the neuroprotective treatment of optic neuritis has been suggested using sodium channel blocking agents, on the basis that such treatment is expected to reduce any energy deficit, such as that resulting from mitochondrial failure [86]. In support of this treatment approach, the sodium channel-blocking agents flecainide, lamotrigine and phenytoin have been proven to provide very effective neuroprotection in models of EAE [87,88,89,90] and experimental autoimmune neuritis [91]. Furthermore, phenytoin has been tested in clinical trial in acute optic neuritis and found to provide very effective neuroprotection, namely, a 30% reduction in the loss of the retinal nerve fibre layer [92]. The beneficial effects of sodium channel blocking agents may not only help to compensate for reduced energy levels but they may also reduce the production of deleterious innate immune factors by their ability to reduce microglial activation [93,94,95,96].

## 4. Materials and Methods

### 4.1. Animals

All experiments were performed in accordance with the UK Home Office Animals (Scientific Procedures) Act (1986). Female Dark Agouti (DA; 140–160 g; Envigo, London, UK) were housed in a 12 h light/dark cycle with food and water ad libitum. EAE (n = 30) was induced using our usual protocol [81,82], which includes subcutaneous injection at the base of the tail with an emulsion containing 100 µL incomplete Freund’s adjuvant (IFA) mixed with 100 µL recombinant MOG (rMOG; a gift from the Christopher Linington lab) under light anaesthesia with isoflurane. Control animals were ‘immunised’ with IFA mixed with 100 µL saline only (n = 15). After immunisation, animals were daily weighed and evaluated for any neurological deficit using a 10-point scale [82], with higher scores indicating more severe deficits. Animals would usually start to show neurological deficits 8–10 days after immunisation, with the deficits reaching a peak approximately 2 days later. The deficits then subsided, manifesting as a remission, followed by a relapse.

Some of the animals were injected intravenously with pimonidazole (60 mg/kg body weight; HPI Inc., Westborough, MA, USA) 24 h prior to trans-cardial perfusion. Under hypoxic tissue conditions (approximately PO_2_ < 10 mmHg), a reduced form of pimonidazole binds permanently to the tissue, which could later be visualised using immunohistological methods [97,98], providing an indication of the absolute oxygen concentration of the tissue. In another batch of animals, dihydroethidium (DHE; Invitrogen, Carlsbad, CA, USA; 1.2 mg/animal) was injected intravenously 4 h prior to trans-cardial perfusion. DHE reacts with superoxide and forms fluorescent labelling [98,99] that can be used as an in vivo guide to the production of superoxide.

### 4.2. Tissue Processing

Animals were trans-cardially perfused with heparinised saline followed by 4% paraformaldehyde (PFA; Sigma, Dorset, UK) either at the peak of the first relapse (day 2 after the onset of neurological deficit), 2 days after the peak (day 4), or at the peak of the second relapse (>20 days post-immunisation). The optic nerves were removed and post-fixed in PFA overnight at 4 °C before being transferred to 30% sucrose (Sigma) for cryoprotection. The optic nerves were embedded in OCT (Thermo-Fisher scientific, Waltham, MA, USA) and cut longitudinally in sections 10 µm thick using a cryostat (Leica, Wetzlar, Germany), and the sections were thaw-mounted onto glass slides (SuperFrost Plus; VWR, Leicestershire, UK).

### 4.3. Histology

Sections were air-dried overnight at room temperature, followed by rehydration with phosphate-buffered saline (PBS; Sigma). The sections were incubated for 1.5 h in a blocking buffer containing 10% normal goat serum (Vector laboratories, Newark, CA, USA) diluted in PBST (PBS with 0.1% Triton X-100). For immunofluorescent labelling, the slides were incubated overnight at 4 °C in related primary antibodies (rabbit-anti-IBA1 1:200, 019-19741, WAKO; mouse-anti-ED1 1:100, MCA341GA, Bio-Rad; rabbit-anti-HIF1a 1:100, ab82832, Abcam; rabbit-anti-pimonidazole 1:200, HP3-1000, HPI Inc.; rabbit-anti-GLUT1 1:100, ab14683, Abcam; rabbit-anti-iNOS 1:100, PA1-036, Invitrogen; mouse-anti-3NT 1:100, ab61392, Abcam) diluted in a blocking buffer and then for 1 h in corresponding secondary antibodies diluted 1:200 (AlexFluor546-conjugated goat-anti-rabbit IgG, A11035; AlexFluor488-conjugated goat-anti-mouse IgG, A11001; Invitrogen), before being mounted with a DAPI-containing mounting medium (Biotium, Fremont, CA, USA). For labelling by immunohistochemistry, biotin-conjugated secondary antibodies (goat-anti-rabbit, BA1000, Vector laboratories) were used, and the slides were incubated in a biotin-streptavidin amplification solution (ABC kit; Vector laboratories) for 1 h, followed by a diaminobenzidine (DAB; Vector laboratories) reaction.

### 4.4. Imaging and Quantification

Immunofluorescent images were acquired with a confocal microscope (LSM710; Zeiss, Oberkochen, Germany) with a 10× objective (Zeiss) and immunohistochemistry images were acquired with a light microscope (Zeiss) under a 40× objective (Nikon, Tokyo, Japan). The microscopy settings were kept constant throughout image acquisition between sections with similar labelling. Image processing was conducted with ImageJ 1.54f software (NIH, Bethesda, MD, USA). For each image, positive labelling was identified as pixels with intensity above a certain threshold, and the relative area of labelling was then quantified as the area of positive labelling divided by the area of the observed tissue. For GLUT1 images, vascular diameter was calculated for each vessel from its area and length. The result generated for each optic nerve was an average of results acquired from several microscopic images taken across the whole length of the optic nerve.

### 4.5. Statistics

Statistical analyses were performed with Excel 2010 (Microsoft, Redmond, WA, USA), Prism 10.0 (GraphPad, La Jolla, CA, USA) or Matlab 2023a (MathWorks, Natick, MA, USA) using Student’s *t*-test, ANOVA or linear regression. The threshold for statistical significance was indicated as not significant (*p* > 0.05), * (*p* ≤ 0.05), ** (*p* ≤ 0.01), *** (*p* ≤ 0.001).

## 5. Conclusions

We conclude that in optic neuritis due to EAE, the acutely inflamed optic nerves are significantly hypoxic. The hypoxia is accompanied by, or may perhaps even be responsible for, the appearance of a range of deleterious innate immune factors, including the formation of reactive oxygen and nitrogen species. The hypoxia and innate immune factors can impair mitochondrial function, causing axonal conduction block and loss of vision, and promoting demyelination. The finding of optic nerve hypoxia in optic neuritis opens new perspectives for the treatment of the disease because preventing hypoxia may not only restore mitochondrial function but also avoid the formation of an environment that promotes demyelination and axonal damage.

## Figures and Tables

**Figure 1 ijms-25-03077-f001:**
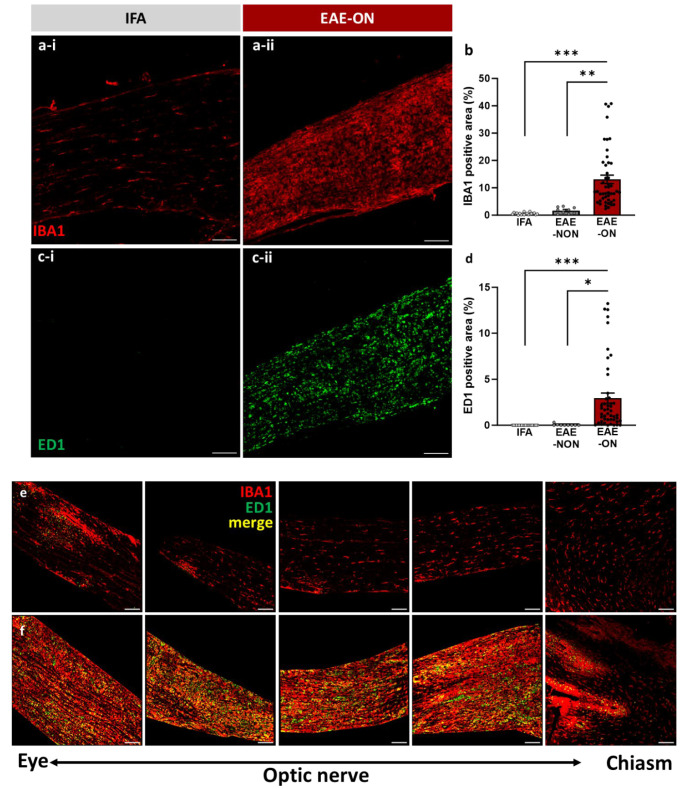
Identification of EAE-ON and EAE-NON. Examples of fluorescent images of optic nerve sections labelled with (**a**) ED1 and (**c**) IBA1 in an IFA animal (IFA; (**a-i**,**c-i**)) and in an inflamed optic nerve of an animal with EAE (EAE-ON; (**a-ii**,**c-ii**)). Both (**b**) ED1 and (**d**) IBA1 showed significantly greater labelling in EAE-ON than in EAE-NON and IFA nerves. Each data point in (**b**,**d**) represents one nerve. (**e**,**f**) show inflamed nerves (EAE-ON). In such nerves, the labelling for ED1 and more intense IBA1 commenced from the orbital end of the nerve (**e**), extending in some nerves towards the chiasm and involving the chiasm (**f**). Mean ± SEM, one-tailed independent *t*-test, * *p* < 0.05, ** *p* < 0.01, *** *p* < 0.001, bar = 100 µm.

**Figure 2 ijms-25-03077-f002:**
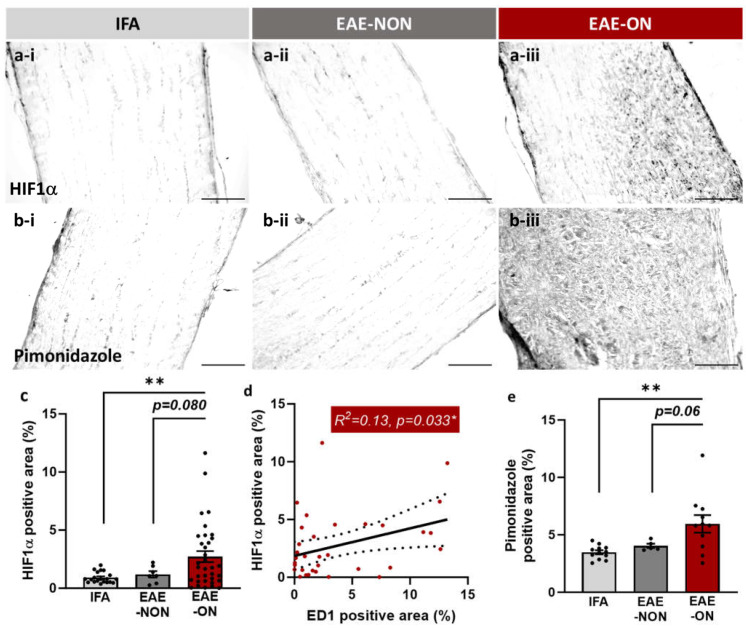
The inflamed optic nerve was hypoxic. Photomicrographs of optic nerve sections labelled for (**a**) HIF1α and (**b**) pimonidazole in tissue from an IFA animal (IFA; (**a-i**,**b-i**)), and tissue representing EAE-NON (**a-ii**,**b-ii**) and EAE-ON (**a-iii**,**b-iii**) from animals immunised for EAE. The EAE-ON nerves showed significantly more intense labelling for (**c**) HIF1α compared with nerves from IFA animals, which was more intense in the most inflamed nerves (**d**). A reduction in absolute tissue oxygenation was confirmed in the EAE-ON nerves by significantly greater labelling for (**e**) pimonidazole compared with nerves from IFA animals. Each data point in (**c**–**e**) represents one nerve. Mean ± SEM, one-tailed independent *t*-test and linear regression (in EAE-ON group; (**d**)), * *p* < 0.05, ** *p* < 0.01, bar = 50 µm.

**Figure 3 ijms-25-03077-f003:**
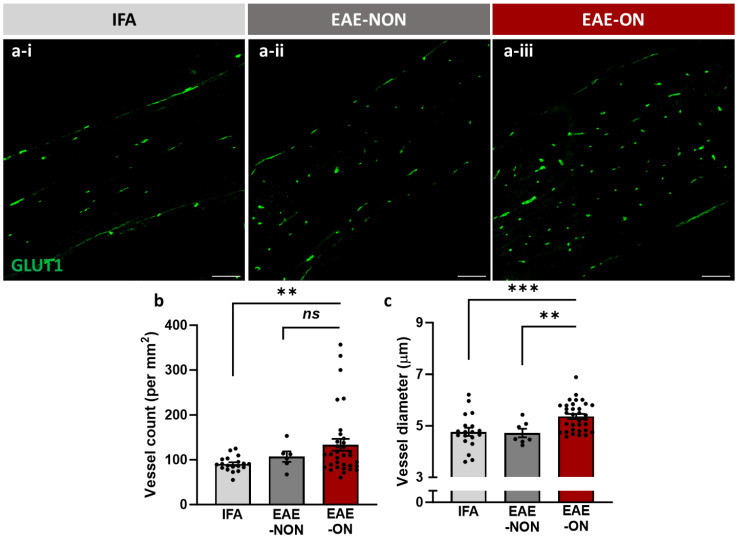
Vascular changes in the inflamed optic nerve. Fluorescent images of sections of optic nerves labelled for (**a**) GLUT1 in nerves from an IFA animal (IFA; (**a-i**)) and nerves identified as EAE-NON (**a-ii**) and EAE-ON (**a-iii**). The inflamed nerve showed a tendency for vessels to be less aligned with the length of the nerve, perhaps due to displacement by inflammatory cells. (**b**) Vessel number appeared significantly increased in the EAE-ON group, accompanied by significant (**c**) dilation compared with the EAE-NON and IFA groups. Each data point in (**b**,**c**) represents one nerve. Mean ± SEM, one-tailed independent *t*-test, ns: not significant, ** *p* < 0.01, *** *p* < 0.001, bar = 100 µm.

**Figure 4 ijms-25-03077-f004:**
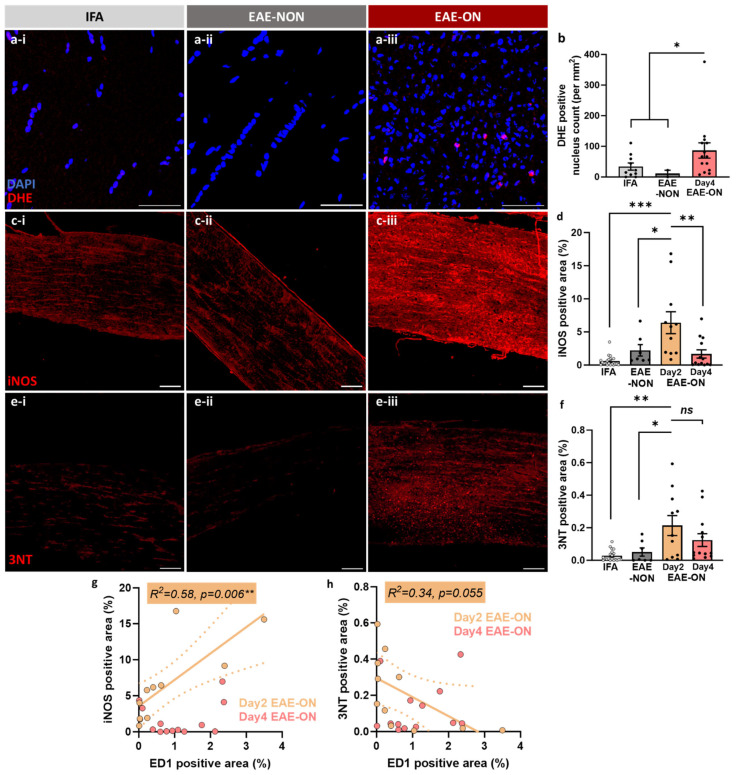
The inflamed optic nerve showed increased labelling for the presence of suspected superoxide (DHE), nitric oxide (iNOS) and peroxynitrite (3NT). Sections of optic nerves showing fluorescence resulting from (**a**) DHE and labelled for (**c**) iNOS; (**e**) 3NT in animals ‘immunised’ with IFA (**a-i**,**c-i**,**e-i**) and those immunised for EAE but without optic neuritis (EAE-NON; (**a-ii**,**c-ii**,**e-ii**)) or with optic neuritis (EAE-ON; (**a-iii**,**c-iii**,**e-iii**)). Inflamed optic nerves from animals with EAE showed significantly more intense fluorescence from (**b**) DHE and labelling for (**b**) iNOS, (**c**) 3NT and (**f**) DHE. The labelling for iNOS was highest for EAE-ON at peak of disease (day 2 after disease onset), but was significantly reduced when examined on day 4 (**d**). On day 2, the labelling for iNOS was positively correlated with the magnitude of inflammation (**g**), but labelling for 3NT was absent in the most inflamed nerves (**h**). Each data point in (**b**,**d**,**f**–**h)** represents one nerve. Mean ± SEM, one-way ANOVA, ns: not significant, * *p* < 0.05, ** *p* < 0.01, *** *p* < 0.001, bar = (**a**,**c**) 100 µm or (**e**,**g**) 50 µm.

**Table 1 ijms-25-03077-t001:** Optic neuritis in animals with EAE in association with hindlimb and tail deficits.

	Hindlimb and Tail Deficits			
	Asymptomatic (Perfusion Day)	First Peak (Day 2 after Disease Onset)	Day 4 after Disease Onset	Second Peak
Optic nerve involvement				
Bilateral	1 (29 days p.i.)	8	6	6
Unilateral	1 (45 days p.i.)	1	0	0
Not inflamed	1 (18 days p.i.)	2	0	0
Only one nerve examined	1 (44 days p.i.; EAE-NON)	0	1 (EAE-ON)	2 (EAE-ON)
Total	4	11	7	8

EAE-ON: nerve from EAE animal showing optic neuritis; EAE-NON: non-inflamed nerve from EAE animal; p.i.: post-immunisation.

## Data Availability

Data contained within the article will be made available upon reasonable request.

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
