# Peer review of "Tissue Hypoxia and Associated Innate Immune Factors in Experimental Autoimmune Optic Neuritis"

_ijms, 2024, doi:10.3390/ijms25053077_

Round 1

Reviewer 1 Report

Comments and Suggestions for Authors

In this study, Yang et al. report the presence of tissue hypoxia and associated markers of oxidative and nitrative stress in the optic nerves of rats with experimental autoimmune optic neuritis (EAE-ON). These findings suggest a potential link between low oxygen levels and the damaging processes observed in the disease. Notably, current treatments for optic neuritis primarily address symptoms but haven't shown success in preventing long-term damage. The authors investigate the presence of hypoxia markers like HIF-1α and pimonidazole, alongside markers of oxidative and nitrative stress, aiming to elucidate the role of hypoxia in EAE-ON and explore potential therapeutic targets.

Below are my critique points:

1. Number and nature of experimental units should be clearly defined for each experiment. This information is crucial for assessing the robustness of the findings. Additionally, treating individual microscopic images from a single animal as separate data points constitutes pseudoreplication and should be avoided.

2. It appears from Figure 2 that the magnification might not be consistent across all micrographs. Furthermore, the quantification of pimonidazole staining does not seem to correspond entirely to the visual impression of the images. For better clarity, the authors could consider demonstrating the threshold used to define positive staining areas.

3. Were different groups of animals were used for correlation analysis?  By pooling data from heterogeneous groups, distinct underlying relationships are masked, leading to a misleading overall correlation. (See: see: https://doi.org/10.1111/1467-9884.00365 Nonsensical and biased correlation due to pooling heterogeneous samples).  Alternative approaches such as separate group analyses or mixed-effects models are crucial to account for potential heterogeneity and provide more robust and interpretable results. 

4. For analyzing count data such as the number of positive cells, employing count regression models is likely more appropriate than the methods currently used.

Reviewer 2 Report

Comments and Suggestions for Authors

This study originates from a group with extensive expertise in the mechanisms of demyelination. Here they assessed whether the optic nerves are hypoxic in the DA rMOG rat model of experimental autoimmune encephalomyelitis (EAE), which is a well-established model in multiple sclerosis research. The principal novel finding is that the inflamed optic nerves labeled significantly for hypoxia-inducible factor-1α (HIF1α) and intravenously administered pimonidazole. 

The conclusion that in optic neuritis due to EAE, the acutely inflamed optic nerves are significantly hypoxic is well supported by the histological data. I only have minor concerns about how the relevance is discussed:

1. I agree with the authors that it is reasonable to suspect that hypoxia of the optic nerve would precipitate the energy crisis and its deleterious effects. Nevertheless, what is the evidence that hypoxia occurs before or independent of the energy crisis? The discussion about the “hypoxic hypoxia” should be expanded.

2. Is the hypoxia at the time of inflammation only deleterious to the optic nerve? This could be especially important to clarify because this study is based on the assumption that the pathology in the spinal cord precedes that in the optic nerves.
